# Lasso Screening Rules via Dual Polytope Projection

**Jie Wang, Jiayu Zhou, Peter Wonka, Jieping Ye**
Computer Science and Engineering
Arizona State University, Tempe, AZ 85287
{jie.wang.ustc, jiayu.zhou, peter.wonka, jieping.ye}@asu.edu

## Abstract

Lasso is a widely used regression technique to find sparse representations. When the dimension of the feature space and the number of samples are extremely large, solving the Lasso problem remains challenging. To improve the efficiency of solving large-scale Lasso problems, El Ghaoui and his colleagues have proposed the SAFE rules which are able to quickly identify the inactive predictors, i.e., predictors that have $0$ components in the solution vector. Then, the inactive predictors or features can be removed from the optimization problem to reduce its scale. By transforming the standard Lasso to its dual form, it can be shown that the inactive predictors include the set of inactive constraints on the optimal dual solution. In this paper, we propose an efficient and effective screening rule via Dual Polytope Projections (DPP), which is mainly based on the uniqueness and nonexpansiveness of the optimal dual solution due to the fact that the feasible set in the dual space is a convex and closed polytope. Moreover, we show that our screening rule can be extended to identify inactive groups in group Lasso. To the best of our knowledge, there is currently no "exact" screening rule for group Lasso. We have evaluated our screening rule using many real data sets. Results show that our rule is more effective in identifying inactive predictors than existing state-of-the-art screening rules for Lasso.

## 1   Introduction

Data with various structures and scales comes from almost every aspect of daily life. To effectively extract patterns in the data and build interpretable models with high prediction accuracy is always desirable. One popular technique to identify important explanatory features is by sparse regularization. For instance, consider the widely used $\ell_1$-regularized least squares regression problem known as Lasso [20]. The most appealing property of Lasso is the sparsity of the solutions, which is equivalent to feature selection. Suppose we have $N$ observations and $p$ predictors. Let $\mathbf{y}$ denote the $N$ dimensional response vector and $\mathbf{X} = [\mathbf{x}_1, \mathbf{x}_2, \ldots, \mathbf{x}_p]$ be the $N \times p$ feature matrix. Let $\lambda \geq 0$ be the regularization parameter, the Lasso problem is formulated as the following optimization problem:

$$\inf_{\beta \in \Re^p} \tfrac{1}{2}\|\mathbf{y} - \mathbf{X}\beta\|_2^2 + \lambda\|\beta\|_1. \tag{1}$$

Lasso has achieved great success in a wide range of applications [5, 4, 28, 3, 23] and in recent years many algorithms have been developed to efficiently solve the Lasso problem [7, 12, 18, 6, 10, 1, 11]. However, when the dimension of feature space and the number of samples are very large, solving the Lasso problem remains challenging because we may not even be able to load the data matrix into main memory. The idea of a screening test proposed by El Ghaoui *et al.* [8] is to first identify inactive predictors that have $0$ components in the solution and then remove them from the optimization. Therefore, we can work on a reduced feature matrix to solve Lasso efficiently.

In [8], the "SAFE" rule discards $\mathbf{x}_i$ when

$$|\mathbf{x}_i^T\mathbf{y}| < \lambda - \|\mathbf{x}_i\|_2\|\mathbf{y}\|_2 \tfrac{\lambda_{max}-\lambda}{\lambda_{max}} \tag{2}$$

where $\lambda_{max} = \max_i |\mathbf{x}_i^T\mathbf{y}|$ is the largest parameter such that the solution is nontrivial. Tibshirani et al. [21] proposed a set of strong rules which were more effective in identifying inactive predictors.

The basic version discards $\mathbf{x}_i$ if $|\mathbf{x}_i^T \mathbf{y}| < 2\lambda - \lambda_{max}$. However, it should be noted that the proposed strong rules might mistakenly discard active predictors, i.e., predictors which have nonzero coefficients in the solution vector. Xiang et al. [26, 25] developed a set of screening tests based on the estimation of the optimal dual solution and they have shown that the SAFE rules are in fact a special case of the general sphere test.

In this paper, we develop new efficient and effective screening rules for the Lasso problem; our screening rules are exact in the sense that no active predictors will be discarded. By transforming problem (1) to its dual form, our motivation is mainly based on three geometric observations in the dual space. First, the active predictors belong to a subset of the active constraints on the optimal dual solution, which is a direct consequence of the KKT conditions. Second, the optimal dual solution is in fact the projection of the scaled response vector onto the feasible set of the dual variables. Third, because the feasible set of the dual variables is closed and convex, the projection is nonexpansive with respect to $\lambda$ [2], which results in an effective estimation of its variation. Moreover, based on the basic DPP rules, we propose the "Enhanced DPP" rules which are able to detect more inactive features than DPP. We evaluate our screening rules on real data sets from many different applications. The experimental results demonstrate that our rules are more effective in discarding inactive features than existing state-of-the-art screening rules.

## 2 Screening Rules for Lasso via Dual Polytope Projections

In this section, we present the basics of the dual formulation of problem (1) including its geometric properties (Section 2.1). Based on the geometric properties of the dual optimal, we develop the fundamental principle in Section 2.2 (Theorem 2), which can be used to construct screening rules for Lasso. In section 2.3, we discuss the relation between dual optimal and LARS [7]. As a straightforward extension of DPP rules, we develop the sequential version of DPP (SDPP) in Section 2.4. Moreover, we present enhanced DPP rules in Section 2.5.

### 2.1 Basics

Different from [26, 25], we do not assume $\mathbf{y}$ and all $\mathbf{x}_i$ have unit length. We first transform problem (1) to its dual form (to make the paper self-contained, we provide the detailed derivation of the dual form in the supplemental materials):

$$\sup_{\theta} \quad \left\{ \tfrac{1}{2}\|\mathbf{y}\|_2^2 - \tfrac{\lambda^2}{2}\|\theta - \tfrac{\mathbf{y}}{\lambda}\|_2^2 : \ |\mathbf{x}_i^T \theta| \le 1, \ i = 1, 2, \ldots, p \right\} \tag{3}$$

where $\theta$ is the dual variable. Since the feasible set, denoted by $F$, is the intersection of $2p$ half-spaces, it is a closed and convex polytope. From the objective function of the dual problem (3), it is easy to see that the optimal dual solution $\theta^*$ is a feasible $\theta$ which is closest to $\frac{\mathbf{y}}{\lambda}$. In other words, $\theta^*$ is the projection of $\frac{\mathbf{y}}{\lambda}$ onto the polytope $F$. Mathematically, for an arbitrary vector $\mathbf{w}$ and a convex set $C$, if we define the projection function as

$$P_C(\mathbf{w}) = \underset{\mathbf{u} \in C}{\operatorname{argmin}} \|\mathbf{u} - \mathbf{w}\|_2, \tag{4}$$

then

$$\theta^* = P_F(\mathbf{y}/\lambda) = \underset{\theta \in F}{\operatorname{argmin}} \left\|\theta - \tfrac{\mathbf{y}}{\lambda}\right\|_2. \tag{5}$$

We know that the optimal primal and dual solutions satisfy:

$$\mathbf{y} = \mathbf{X}\beta^* + \lambda\theta^* \tag{6}$$

and the KKT conditions for the Lasso problem (1) are

$$(\theta^*)^T \mathbf{x}_i \in \begin{cases} \operatorname{sign}([\beta^*]_i) \text{ if } [\beta^*]_i \neq 0 \\ [-1, 1] \text{ if } [\beta^*]_i = 0 \end{cases} \tag{7}$$

where $[\cdot]_k$ denotes the $k^{th}$ component.

By the KKT conditions in Eq. (7), if the inner product $(\theta^*)^T \mathbf{x}_i$ belongs to the open interval $(-1, 1)$, then the corresponding component $[\beta^*]_i$ in the solution vector $\beta^*(\lambda)$ has to be 0. As a result, $\mathbf{x}_i$ is an inactive predictor and can be removed from the optimization.

On the other hand, let $\partial H(\mathbf{x}_i) = \{\mathbf{z}: \mathbf{z}^T \mathbf{x}_i = 1\}$ and $H(\mathbf{x}_i)_- = \{\mathbf{z}: \mathbf{z}^T \mathbf{x}_i \le 1\}$ be the hyperplane and half space determined by $\mathbf{x}_i$ respectively. Consider the dual problem (3); constraints induced by each $\mathbf{x}_i$ are equivalent to requiring each feasible $\theta$ to lie inside the intersection of $H(\mathbf{x}_i)_-$ and $H(-\mathbf{x}_i)_-$. If $|(\theta^*)^T \mathbf{x}_i| = 1$, i.e., either $\theta^* \in \partial H(\mathbf{x}_i)_-$ or $\theta^* \in \partial H(-\mathbf{x}_i)_-$, we say the constraints induced by $\mathbf{x}_i$ are active on $\theta^*$.

We define the "*active*" set on $\theta^*$ as $\mathcal{I}_{\theta^*} = \{i\colon |(\theta^*)^T\mathbf{x}_i| = 1, i \in \mathcal{I}\}$ where $\mathcal{I} = \{1, 2, \ldots, p\}$. Otherwise, if $\theta^*$ lies between $\partial H(\mathbf{x}_i)$ and $\partial H(-\mathbf{x}_i)$, i.e., $|(\theta^*)^T\mathbf{x}_i| < 1$, we can safely remove $\mathbf{x}_i$ from the problem because $[\beta^*]_i = 0$ according to the KKT conditions in Eq. (7). Similarly, the "*inactive*" set on $\theta^*$ is defined as $\overline{\mathcal{I}}_{\theta^*} = \mathcal{I} \setminus \mathcal{I}_{\theta^*}$. Therefore, from a geometric perspective, if we know $\theta^*$, i.e., the projection of $\frac{\mathbf{y}}{\lambda}$ onto $F$, the predictors in the inactive set on $\theta^*$ can be discarded from the optimization. It is worthwhile to mention that inactive predictors, i.e., predictors that have 0 components in the solution, are not the same as predictors in the inactive set. In fact, by the KKT conditions, predictors in the inactive set must be inactive predictors since they are guaranteed to have 0 components in the solution, but the converse may not be true.

## 2.2 Fundamental Screening Rules via Dual Polytope Projections

Motivated by the above geometric intuitions, we next show how to find the predictors in the inactive set on $\theta^*$. To emphasize the dependence on $\lambda$, let us write $\theta^*(\lambda)$ and $\beta^*(\lambda)$. If we know exactly where $\theta^*(\lambda)$ is, it will be trivial to find the predictors in the inactive set. Unfortunately, in most of the cases, we only have incomplete information about $\theta^*(\lambda)$ without actually solving problem (1) or (3). Suppose we know the exact $\theta^*(\lambda')$ for a specific $\lambda'$. How can we estimate $\theta^*(\lambda'')$ for another $\lambda''$ and its inactive set? To answer this question, we start from Eq. (5); $\theta^*(\lambda)$ is nonexpansive because it is a projection operator. For convenience, we cite the projection theorem in [2] as follows.

**Theorem 1.** *Let $C$ be a convex set, then the projection function defined in Eq. (4) is continuous and nonexpansive, i.e.,*

$$\|P_C(\mathbf{w}_2) - P_C(\mathbf{w}_1)\|_2 \leq \|\mathbf{w}_2 - \mathbf{w}_1\|_2, \ \forall \mathbf{w}_2, \mathbf{w}_1. \tag{8}$$

Given $\theta^*(\lambda')$, the next theorem shows how to estimate $\theta^*(\lambda'')$ and its inactive set for another parameter $\lambda''$.

**Theorem 2.** *For the Lasso problem, assume we are given the solution of its dual problem $\theta^*(\lambda')$ for a specific $\lambda'$. Let $\lambda''$ be a nonnegative value different from $\lambda'$. Then $[\beta^*(\lambda'')]_i = 0$ if*

$$|\mathbf{x}_i^T\theta^*(\lambda')| < 1 - \|\mathbf{x}_i\|_2\|\mathbf{y}\|_2\left|\frac{1}{\lambda'} - \frac{1}{\lambda''}\right|. \tag{9}$$

*Proof.* From the KKT conditions in Eq. (7), we know $|\mathbf{x}_i^T\theta^*(\lambda'')| < 1 \Rightarrow [\beta^*(\lambda'')]_i = 0$. By the dual problem (3), $\theta^*(\lambda)$ is the projection of $\frac{\mathbf{y}}{\lambda}$ onto the feasible set $F$. According to the projection theorem [2], that is, Theorem 1, for closed convex sets, $\theta^*(\lambda)$ is continuous and nonexpansive, i.e.,

$$\|\theta^*(\lambda'') - \theta^*(\lambda')\|_2 \leq \left\|\frac{\mathbf{y}}{\lambda''} - \frac{\mathbf{y}}{\lambda'}\right\|_2 = \|\mathbf{y}\|_2\left|\frac{1}{\lambda''} - \frac{1}{\lambda'}\right| \tag{10}$$

Then

$$|\mathbf{x}_i^T\theta^*(\lambda'')| \leq |\mathbf{x}_i^T\theta^*(\lambda'') - \mathbf{x}_i^T\theta^*(\lambda')| + |\mathbf{x}_i^T\theta^*(\lambda')| \tag{11}$$
$$< \|\mathbf{x}_i\|_2\|(\theta^*(\lambda'') - \theta^*(\lambda'))\|_2 + 1 - \|\mathbf{x}_i\|_2\|\mathbf{y}\|_2\left|\frac{1}{\lambda''} - \frac{1}{\lambda'}\right|$$
$$\leq \|\mathbf{x}_i\|_2\|\mathbf{y}\|_2\left|\frac{1}{\lambda''} - \frac{1}{\lambda'}\right| + 1 - \|\mathbf{x}_i\|_2\|\mathbf{y}\|_2\left|\frac{1}{\lambda''} - \frac{1}{\lambda'}\right| = 1$$

which completes the proof. $\qquad\square$

From theorem 2, it is easy to see our rule is quite flexible since every $\theta^*(\lambda')$ would result in a new screening rule. And the smaller the gap between $\lambda'$ and $\lambda''$, the more effective the screening rule is. By "*more effective*", we mean a stronger capability of identifying inactive predictors.

As an example, let us find out $\theta^*(\lambda_{max})$. Recall that $\lambda_{max} = \max_i |\mathbf{x}_i^T\mathbf{y}|$. It is easy to verify $\frac{\mathbf{y}}{\lambda_{max}}$ is itself feasible. Therefore the projection of $\frac{\mathbf{y}}{\lambda_{max}}$ onto $F$ is itself, i.e., $\theta^*(\lambda_{max}) = \frac{\mathbf{y}}{\lambda_{max}}$. Moreover, by noting that for $\forall \lambda > \lambda_{max}$, we have $|\mathbf{x}_i^T\mathbf{y}/\lambda| < 1, i \in \mathcal{I}$, i.e., all predictors are in the inactive set at $\theta^*(\lambda)$, we conclude that the solution to problem (1) is 0. Combining all these together and plugging $\theta^*(\lambda_{max}) = \frac{\mathbf{y}}{\lambda_{max}}$ into Eq. (9), we obtain the following screening rule.

**Corollary 3. DPP**: *For the Lasso problem (1), let $\lambda_{max} = \max_i |\mathbf{x}_i^T\mathbf{y}|$. If $\lambda \geq \lambda_{max}$, then $[\beta^*]_i = 0, \forall i \in \mathcal{I}$. Otherwise, $[\beta^*(\lambda)]_i = 0$ if*

$$\left|\mathbf{x}_i^T\frac{\mathbf{y}}{\lambda_{max}}\right| < 1 - \|\mathbf{x}_i\|_2\|\mathbf{y}\|_2\left(\frac{1}{\lambda} - \frac{1}{\lambda_{max}}\right).$$

Clearly, DPP is most effective when $\lambda$ is close to $\lambda_{max}$. So how can we find a new $\theta^*(\lambda')$ with $\lambda' < \lambda_{max}$? Note that Eq. (6) is in fact a natural bridge which relates the primal and dual optimal solutions. As long as we know $\beta^*(\lambda')$, it is easy to get $\theta^*(\lambda')$ when $\lambda$ is relatively small, e.g., LARS [7] and Homotopy [17] algorithms.

Table 1: Illustration of the running time for DPP screening and for solving the Lasso problem after screening. $T_s$: time for screening. $T_l$: time for solving the Lasso problem after screening. $T_o$: the total time. Entries of the response vector $\mathbf{y}$ are i.i.d. by a standard Gaussian. Columns of the data matrix $\mathbf{X} \in \Re^{1000 \times 100000}$ are generated by $\mathbf{x}_i = \mathbf{y} + \alpha\mathbf{z}$ where $\alpha$ is a random number drawn uniformly from $[0, 1]$. Entries of $\mathbf{z}$ are i.i.d. by a standard Gaussian. $\lambda_{max} = 0.95$ and $\lambda/\lambda_{max}$=0.5.

|         | LASSO   | DPP    | DPP2   | DPP5   | DPP10  | DPP20  |
|---------|---------|--------|--------|--------|--------|--------|
| $T_s$ (S) | —       | 0.035  | 0.073  | 0.152  | 0.321  | 0.648  |
| $T_l$ (S) | —       | 10.250 | 9.634  | 8.399  | 1.369  | 0.121  |
| $T_o$ (S) | 103.314 | 10.285 | 9.707  | 8.552  | 1.690  | 0.769  |

**Remark**: Xiang *et al.* [26] developed a general sphere test which says that if $\theta^*$ is estimated to be inside a ball $\|\theta^* - \mathbf{q}\|_2 \leq r$, then $|\mathbf{x}_i^T\mathbf{q}| < (1 - r) \Rightarrow [\beta^*]_i = 0$. Considering the DPP rules in Theorem 2, it is equivalent to setting $\mathbf{q} = \theta^*(\lambda')$ and $r = |\frac{1}{\lambda'} - \frac{1}{\lambda''}|$. Therefore, different from the sphere test and Dome developed in [26, 25] with the radius $r$ fixed at the beginning, the construction of our DPP rules is equivalent to an "$r$" decreasing process. Clearly, the smaller $r$ is, the more effective the DPP rules will be.

**Remark**: Notice that, DPP is not the same as ST1 [26] and SAFE [8], which discards the $i^{th}$ feature if $|\mathbf{x}_i^T\mathbf{y}| < \lambda - \|\mathbf{x}_i\|_2\|\mathbf{y}\|_2\frac{\lambda_{max}-\lambda}{\lambda_{max}}$. From the perspective of the sphere test, the radius of ST1/SAFE and DPP are the same. But the centers of ST1 and DPP are $\mathbf{y}/\lambda$ and $\mathbf{y}/\lambda_{max}$ respectively, which leads to different formulas, i.e., Eq. (2) and Corollary 3.

## 2.3 DPP Rules with LARS/Homotopy Algorithms

It is well known that under mild conditions, the set $\{\beta^*(\lambda) : \lambda > 0\}$ (also know as regularization path [15]) is continuous piecewise linear [17, 7, 15]. The output of LARS or Homotopy algorithms is in fact a sequence of values like $(\beta^*(\lambda^{(0)}), \lambda^{(0)}), (\beta^*(\lambda^{(1)}), \lambda^{(1)}), \ldots$, where $\beta^*(\lambda^{(i)})$ corresponds to the $i$th breakpoint of the regularization path $\{\beta^*(\lambda) : \lambda > 0\}$ and $\lambda^{(i)}$s are monotonically decreasing. By Eq. (6), once we get $\beta^*(\lambda^{(i)})$, we can immediately compute $\theta^*(\lambda^{(i)})$. Then according to Theorem 2, we can construct a DPP rule based on $\theta^*(\lambda^{(i)})$ and $\lambda^{(i)}$. For convenience, if the DPP rule is built based on $\theta^*(\lambda^{(i)})$, we add the index $i$ as suffix to DPP, e.g., DPP5 means it is developed based on $\theta^*(\lambda^{(5)})$. It should be noted that LARS or Homotopy algorithms are very efficient to find the first few breakpoints of the regularization path and the corresponding parameters. For the first few breakpoints, the computational cost is roughly $O(Np)$, i.e., linear with the size of the data matrix $\mathbf{X}$. In Table 1, we report both the time used for screening and the time needed to solve the Lasso problem after screening. The Lasso solver is from the SLEP [14] package.

From Table 1, we can see that compared with the time saved by the screening rules, the time used for screening is negligible. The efficiency of the Lasso solver is improved by DPP20 more than 130 times. In practice, DPP rules built on the first few $\theta^*(\lambda^{(i)})$'s lead to more significant performance improvement than existing state-of-art screening tests. We will demonstrate the effectiveness of our DPP rules in the experiment section. As another useful property of LARS/Homotopy algorithms, it is worthwhile to mention that changes of the active set only happen at the breakpoints [17, 7, 15]. Consequently, given the parameters corresponding to a pair of adjacent breakpoints, e.g., $\lambda^{(i)}$ and $\lambda^{(i+1)}$, the active set for $\lambda \in (\lambda^{(i+1)}, \lambda^{(i)})$ is the same as $\lambda = \lambda^{(i)}$. Therefore, besides the sequence of breakpoints and the associated parameters $(\beta^*(\lambda^{(0)}), \lambda^{(0)}), \ldots (\beta^*(\lambda^{(k)}), \lambda^{(k)})$ computed by LARS/Homotopy algorithms, we know the active set for $\forall\lambda \geq \lambda^{(k)}$. Hence we can remove the predictors in the inactive set from the optimization problem (1). This scheme has been embedded in DPP rules.

**Remark**: Some works, e.g., [21], [8], solve several Lasso problems for different parameters to improve the screening performance. However, the DPP algorithms do not aim to solve a sequence of Lasso problems, but just to accelerate one. The LARS/Homotopy algorithms are used to find the first few breakpoints of the regularization path and the corresponding parameters, instead of solving general Lasso problems. Thus, different from [21], [8] who need to iteratively compute a screening step and a Lasso step, DPP algorithms only compute one screening step and one Lasso step.

## 2.4 Sequential Version of DPP Rules

Motivated by the ideas of [21] and [8], we can develop a sequential version of DPP rules. In other words, if we are given a sequence of parameter values $\lambda_1 > \lambda_2 > \ldots > \lambda_m$, we can first apply DPP to discard inactive predictors for the Lasso problem (1) with parameter being $\lambda_1$. After solving

the reduced optimization problem for $\lambda_1$, we obtain the exact solution $\beta^*(\lambda_1)$. Hence by Eq. (6), we can find $\theta^*(\lambda_1)$. According to Theorem 2, once we know the optimal dual solution $\theta^*(\lambda_1)$, we can construct a new screening rule to identify inactive predictors for problem (1) with $\lambda = \lambda_2$. By repeating the above process, we obtain the sequential version of the DPP rule (SDPP).

**Corollary 4. SDPP**: *For the Lasso problem (1), suppose we are given a sequence of parameter values* $\lambda_{max} = \lambda_0 > \lambda_1 > \ldots > \lambda_m$. *Then for any integer* $0 \le k < m$, *we have* $[\beta^*(\lambda_{k+1})]_i = 0$ *if* $\beta^*(\lambda_k)$ *is known and the following holds:*

$$\left| \mathbf{x}_i^T \frac{\mathbf{y} - \mathbf{X}\beta^*(\lambda_k)}{\lambda_k} \right| < 1 - \|\mathbf{x}_i\|_2 \|\mathbf{y}\|_2 \left( \frac{1}{\lambda_{k+1}} - \frac{1}{\lambda_k} \right).$$

**Remark**: There are some other related works on screening rules, e.g., Wu *et al.* [24] built screening rules for $\ell_1$ penalized logistic regression based on the inner products between the response vector and each predictor; Tibshirani *et al.* [21] developed strong rules for a set of Lasso-type problems via the inner products between the residual and predictors; in [9], Fan and Lv studied screening rules for Lasso and related problems. But all of the above works may mistakenly discard predictors that have non-zero coefficients in the solution. Similar to [8, 26, 25], our DPP rules are exact in the sense that the predictors discarded by our rules are inactive predictors, i.e., predictors that have zero coefficients in the solution.

## 2.5 Enhanced DPP Rules

In this section, we show how to further improve the DPP rules. From the inequality in (9), we can see that the larger the right hand side is, the more inactive features can be detected. From the proof of Theorem 2, we need to make the right hand side of the inequality in (10) as small as possible. By noting that $\theta^*(\lambda') = P_F(\frac{\mathbf{y}}{\lambda'})$ and $\theta^*(\lambda'') = P_F(\frac{\mathbf{y}}{\lambda''})$ [please refer to Eq. (5)], the inequality in (10) is in fact a direct consequence of Theorem 1 by letting $C := F$, $\mathbf{w}_1 := \frac{\mathbf{y}}{\lambda'}$ and $\mathbf{w}_2 := \frac{\mathbf{y}}{\lambda''}$.

On the other hand, suppose $\frac{\mathbf{y}}{\lambda'} \notin F$, i.e., $\lambda' \in (0, \lambda_{max})$. It is clear that $\frac{\mathbf{y}}{\lambda'} \ne P_F(\frac{\mathbf{y}}{\lambda'}) = \theta^*(\lambda')$. Let $\theta(t) = \theta^*(\lambda') + t(\frac{\mathbf{y}}{\lambda'} - \theta^*(\lambda'))$ for $t \ge 0$, i.e., $\theta(t)$ is a point lying on the ray starting from $\theta^*(\lambda')$ and pointing to the same direction as $\frac{\mathbf{y}}{\lambda'} - \theta^*(\lambda')$. We can observe that $P_F(\theta(t)) = \theta^*(\lambda')$, i.e., the projection of $\theta(t)$ onto the set $F$ is $\theta^*(\lambda')$ as well (please refer to Lemma A in the supplement for details). By applying Theorem 1 again, we have

$$\|\theta^*(\lambda'') - \theta^*(\lambda')\|_2 = \|P_F(\frac{\mathbf{y}}{\lambda''}) - P_F(\theta(t))\|_2 \le \|\frac{\mathbf{y}}{\lambda''} - \theta(t)\|_2 = \|t(\frac{\mathbf{y}}{\lambda'} - \theta^*(\lambda')) - (\frac{\mathbf{y}}{\lambda''} - \theta^*(\lambda'))\|_2. \tag{12}$$

Clearly, when $t = 1$, the inequality in (12) reduces to the one in (10). Because the inequality in (12) holds for all $t \ge 0$, we may get a tighter bound by

$$\|\theta^*(\lambda'') - \theta^*(\lambda')\|_2 \le \min_{t \ge 0} \|t\mathbf{v}_1 - \mathbf{v}_2\|_2, \tag{13}$$

where $\mathbf{v}_1 = \frac{\mathbf{y}}{\lambda'} - \theta^*(\lambda')$ and $\mathbf{v}_2 = \frac{\mathbf{y}}{\lambda''} - \theta^*(\lambda')$. When $\lambda' = \lambda_{max}$, we can set $\mathbf{v}_1 = \text{sign}(\mathbf{x}_*^T \mathbf{y})\mathbf{x}_*$ where $\mathbf{x}_* := \text{argmax}_{\mathbf{x}_i} |\mathbf{x}_i^T \mathbf{y}|$ (please refer to Lemma B in the supplement for details). The minimization problem on the right hand side of the inequality (13) can be easily solved as follows:

$$\min_{t \ge 0} \|t\mathbf{v}_1 - \mathbf{v}_2\|_2 = \varphi(\lambda', \lambda'') = \begin{cases} \|\mathbf{v}_2\|_2, & \text{if } \langle \mathbf{v}_1, \mathbf{v}_2 \rangle < 0, \\ \left\| \mathbf{v}_2 - \frac{\langle \mathbf{v}_1, \mathbf{v}_2 \rangle}{\|\mathbf{v}_1\|_2^2} \mathbf{v}_1 \right\|_2, & \text{otherwise.} \end{cases} \tag{14}$$

Similar to Theorem 2, we have the following result:

**Theorem 5.** *For the Lasso problem, assume we are given the solution of its dual problem* $\theta^*(\lambda')$ *for a specific* $\lambda'$. *Let* $\lambda''$ *be a nonnegative value different from* $\lambda'$. *Then* $[\beta^*(\lambda'')]_i = 0$ *if*

$$|\mathbf{x}_i^T \theta^*(\lambda')| < 1 - \|\mathbf{x}_i\|_2 \varphi(\lambda', \lambda''). \tag{15}$$

As we explained above, the right hand side of the inequality (15) is no less than that of the inequality (9). Thus, the enhanced DPP is able to detect more inactive features than DPP. The analogues of Corollaries 3 and 4 can be easily derived as well.

**Corollary 6. DPP***: *For the Lasso problem (1), let* $\lambda_{max} = \max_i |\mathbf{x}_i^T \mathbf{y}|$. *If* $\lambda \ge \lambda_{max}$, *then* $[\beta^*]_i = 0, \forall i \in \mathcal{I}$. *Otherwise,* $[\beta^*(\lambda)]_i = 0$ *if the following holds:*

$$\left| \mathbf{x}_i^T \frac{\mathbf{y}}{\lambda_{max}} \right| < 1 - \|\mathbf{x}_i\|_2 \varphi(\lambda_{max}, \lambda).$$

**Corollary 7. SDPP***: *For the Lasso problem (1), suppose we are given a sequence of parameter values* $\lambda_{max} = \lambda_0 > \lambda_1 > \ldots > \lambda_m$. *Then for any integer* $0 \le k < m$, *we have* $[\beta^*(\lambda_{k+1})]_i = 0$

*if $\beta^*(\lambda_k)$ is known and the following holds:*

$$\left| \mathbf{x}_i^T \frac{\mathbf{y} - \mathbf{X}\beta^*(\lambda_k)}{\lambda_k} \right| < 1 - \|\mathbf{x}_i\|_2 \varphi(\lambda_k, \lambda_{k+1}).$$

To simplify notations, we denote the enhanced DPP and SDPP by DPP* and SDPP* respectively.

## 3 Extensions to Group Lasso

To demonstrate the flexibility of DPP rules, we extend our idea to the group Lasso problem [27]:

$$\inf_{\beta \in \Re^p} \frac{1}{2} \|\mathbf{y} - \sum_{g=1}^G \mathbf{X}_g \beta_g\|_2^2 + \lambda \sum_{g=1}^G \sqrt{n_g} \|\beta_g\|_2, \tag{16}$$

where $\mathbf{X}_g \in \Re^{N \times n_g}$ is the data matrix for the $g$th group and $p = \sum_{g=1}^G n_g$. The corresponding dual problem of (16) is (see detailed derivation in the supplemental materials):

$$\sup_{\theta} \quad \left\{ \frac{1}{2} \|\mathbf{y}\|_2^2 - \frac{\lambda^2}{2} \|\theta - \frac{\mathbf{y}}{\lambda}\|_2^2 : \|\mathbf{X}_g^T \theta\|_2 \leq \sqrt{n_g}, g = 1, 2, \ldots, G \right\} \tag{17}$$

Similar to the Lasso problem, the primal and dual optimal solutions of the group Lasso satisfy:

$$\mathbf{y} = \sum_{g=1}^G \mathbf{X}_g \beta_g^* + \lambda \theta^* \tag{18}$$

and the KKT conditions are:

$$(\theta^*)^T \mathbf{X}_g \in \begin{cases} \sqrt{n_g} \frac{\beta_g^*}{\|\beta_g^*\|_2} & \text{if } \beta_g^* \neq 0 \\ \sqrt{n_g} \mathbf{u}, \|\mathbf{u}\|_2 \leq 1 & \text{if } \beta_g^* = 0 \end{cases} \tag{19}$$

for $g = 1, 2, \ldots, G$. Clearly, if $\|(\theta^*)^T \mathbf{X}_g\|_2 < \sqrt{n_g}$, we can conclude that $\beta_g^* = 0$.

Consider problem (17). It is easy to see that the dual optimal $\theta^*$ is the projection of $\frac{\mathbf{y}}{\lambda}$ onto the feasible set. For each $g$, the constraint $\|\mathbf{X}_g^T \theta\|_2 \leq \sqrt{n_g}$ confines $\theta$ to an ellipsoid which is closed and convex. Therefore, the feasible set of the dual problem (17) is the intersection of ellipsoids and thus closed and convex. Hence $\theta^*(\lambda)$ is also nonexpansive for the group lasso problem. Similar to Theorem 2, we can readily develop the following theorem for group Lasso.

**Theorem 8.** *For the group Lasso problem, assume we are given the solution of its dual problem $\theta^*(\lambda')$ for a specific $\lambda'$. Let $\lambda''$ be a nonnegative value different from $\lambda'$. Then $\beta_g^*(\lambda'') = 0$ if*

$$\|\mathbf{X}_g^T \theta^*(\lambda')\|_2 < \sqrt{n_g} - \|\mathbf{X}_g\|_F \|\mathbf{y}\|_2 \left| \frac{1}{\lambda'} - \frac{1}{\lambda''} \right| \tag{20}$$

Similar to the Lasso problem, let $\lambda_{max} = \max_g \|\mathbf{X}_g^T \mathbf{y}\|_2 / \sqrt{n_g}$, we can see that $\frac{\mathbf{y}}{\lambda_{max}}$ is itself feasible, and $\lambda_{max}$ is the largest parameter such that problem (16) has a nonzero solution. Clearly, $\theta^*(\lambda_{max}) = \frac{\mathbf{y}}{\lambda_{max}}$. Similar to DPP and SDPP, we can construct GDPP and SGDPP for group Lasso.

**Corollary 9. GDPP**: *For the group Lasso problem (16), let $\lambda_{max} = \max_g \|\mathbf{X}_g^T \mathbf{y}\|_2 / \sqrt{n_g}$. If $\lambda \geq \lambda_{max}$, $\beta_g^*(\lambda) = 0, \forall g = 1, 2, \ldots, G$. Otherwise, we have $\beta_g^*(\lambda) = 0$ if the following holds:*

$$\left\| \mathbf{X}_g^T \frac{\mathbf{y}}{\lambda_{max}} \right\|_2 < \sqrt{n_g} - \|\mathbf{X}_g\|_F \|\mathbf{y}\|_2 \left( \frac{1}{\lambda} - \frac{1}{\lambda_{max}} \right). \tag{21}$$

**Corollary 10. SGDPP**: *For the group Lasso problem (16), suppose we are given a sequence of parameter values $\lambda_{max} = \lambda_0 > \lambda_1 > \ldots > \lambda_m$. For any integer $0 \leq k < m$, we have $\beta_g^*(\lambda_{k+1}) = 0$ if $\beta^*(\lambda_k)$ is known and the following holds:*

$$\left\| \mathbf{X}_g^T \frac{\mathbf{y} - \sum_{g=1}^G \mathbf{X}_g \beta_g^*(\lambda_k)}{\lambda_k} \right\|_2 < \sqrt{n_g} - \|\mathbf{X}_g\|_F \|\mathbf{y}\|_2 \left( \frac{1}{\lambda_{k+1}} - \frac{1}{\lambda_k} \right). \tag{22}$$

**Remark**: Similar to DPP*, we can develop the enhanced GDPP by simply replacing the term $\|\mathbf{y}\|_2(1/\lambda - 1/\lambda_{max})$ on the right hand side of the inequality (21) with $\varphi(\lambda_{max}, \lambda)$. Notice that, to compute $\varphi(\lambda_{max}, \lambda)$, we set $\mathbf{v}_1 = \mathbf{X}_*(\mathbf{X}_*)^T \mathbf{y}$ where $\mathbf{X}_* = \operatorname{argmax}_{\mathbf{X}_g} \|\mathbf{X}_g^T \mathbf{y}\|_2 / \sqrt{n_g}$ (please refer to Lemma C in the supplement for details). The analogs of SDPP*, that is, SGDPP*, can be obtained by replacing the term $\|\mathbf{y}\|_2(1/\lambda_{k+1} - 1/\lambda_k)$ on the right hand side of the inequality (22) with $\varphi(\lambda_k, \lambda_{k+1})$.

## 4 Experiments

In section 4.1, we first evaluate the DPP and DPP* rules on both real and synthetic data. We then compare the performance of DPP with Dome (see [25, 26]) which achieves state-of-art performance for the Lasso problem among exact screening rules [25]. We evaluate GDPP and SGDPP for the group Lasso problem on three synthetic data sets in section 4.2. We are not aware of any "*exact*" screening rules for the group Lasso problem at this point.

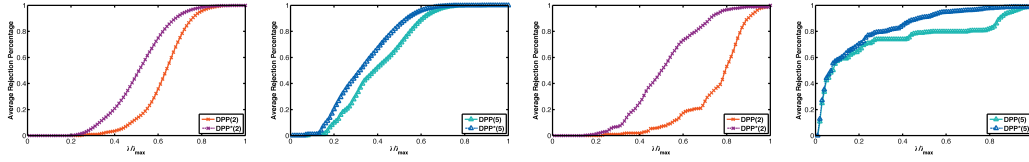

| (a) MNIST-DPP2/DPP$^*$2 | (b) MNIST-DPP5/DPP$^*$5 | (c) COIL-DPP2/DPP$^*$2 | (d) COIL-DPP5/DPP$^*$5 |

Figure 1: Comparison of DPP and DPP$^*$ rules on the MNIST and COIL data sets.

To measure the performance of our screening rules, we compute the rejection rate, i.e., the ratio between the number of predictors discarded by screening rules and the actual number of zero predictors in the ground truth. Because the DPP rules are exact, i.e., no active predictors will be mistakenly discarded, the rejection rate will be less than one. For SAFE and Dome, it is not straightforward to extend them to the group Lasso problem. Similarly to previous works [26], we do not report the computational time saved by screening because it can be easily computed from the rejection ratio. Specifically, if the Lasso solver is linear in terms of the size of the data matrix $\mathbf{X}$, a $K\%$ rejection of the data can save $K\%$ computational time. The general experiment settings are as follows. For each data set, after we construct the data matrix $\mathbf{X}$ and the response $\mathbf{y}$, we run the screening rules along a sequence of 100 values equally spaced on the $\lambda/\lambda_{max}$ scale from 0 to 1. We repeat the procedure 100 times and report the average performance at each of the 100 values of $\lambda/\lambda_{max}$. All of the screening rules are implemented in Matlab. The experiments are carried out on a Intel(R) (i7-2600) 3.4Ghz processor.

### 4.1 DPPs and DPP$^*$s for the Lasso Problem

In this experiment, we first compare the performance of the proposed DPP rules with the enhanced DPP rules (DPP$^*$) on (**a**) the MNIST handwritten digit data set [13]; (**b**) the COIL rotational image data set [16] in Section 4.1.1. We show that the DPP$^*$ rules are more effective in identifying inactive features than the DPP rules. This demonstrate our theoretical results in Section 2.5. Then we evaluate the DPP$^*$/SDPP$^*$ rules and Dome on (**c**) the ADNI data set; (**d**) the Olivetti Faces data set [19]; (**e**) Yahoo web pages data sets [22] and (**f**) a synthetic data set whose entries are i.i.d. by a standard Gaussian.

### 4.1.1 Comparison of DPP and DPP$^*$

As we explain in Section 2.5, all inactive feature detected by the DPP rules can also be detected by the DPP$^*$ rules. But conversely, it is not necessarily true. To demonstrate the advantage of the DPP$^*$ rules, we run DPP2, DPP$^*$2, DPP5 and DPP$^*$5 on the MNIST and COIL data sets. **a)** The MNIST data set contains grey images of scanned handwritten digits, including $60,000$ for training and $10,000$ for testing. The dimension of each image is $28 \times 28$. Each time, we first randomly select 100 images for each digit (and in total we have 1000 images) and get a data matrix $\mathbf{X} \in \Re^{784 \times 1000}$. Then we randomly pick an image as the response $\mathbf{y} \in \Re^{784}$. **b)** The COIL data set includes 100 objects, each of which has 72 color images with $128 \times 128$ pixels. The images that belong to the same object are taken every 5 degree by rotating the object. We use the images of object 10. Each time, we randomly pick one of the images as the response vector $\mathbf{y} \in \Re^{49152}$ and use all the remaining ones to construct the data matrix $\mathbf{X} \in \Re^{49152 \times 71}$. The average $\lambda_{max}$ for the so cultured MNIST and the COIL data sets are $0.837$ and $0.986$. Clearly, the predictors in the data sets are high correlated.

From Figure 1, we observe that DPP$^*$2 significantly outperforms DPP2 for both data sets, especially when $\lambda/\lambda_{max}$ is small. We also observe the same pattern for DPP5 and DPP$^*$5, verifying the claims about DPP$^*$ made in the paper. Thus, in the following experiments, we only report the performance of DPP$^*$ and the competing algorithm Dome.

### 4.1.2 Comparison of DPP$^*$/SDPP$^*$ and Dome

In this experiment, we compare DPP$^*$/SDPP$^*$ rules with Dome. We only report the performance of DPP$^*$5 and DPP$^*$10 among the family of DPP$^*$ rules on the following four data sets.

**c)** The Alzheimer's disease neuroimaging initiative (ADNI; available at www.loni.ucla.edu/ADNI) studies the disease progression of Alzheimer's. The ADNI data set includes 434 patients with 306 features extracted from their baseline MRI scans. Each time we randomly select $90\%$ samples to construct the data matrix $\mathbf{X} \in \Re^{391 \times 306}$. The response $\mathbf{y}$ is the patients' MMSE cognitive scores [29]. **d)** The Olivetti faces data set includes 400 grey scale face images of size $64 \times 64$ for 40 people (10 for each). Each time, we randomly take one of the images as the response vector $\mathbf{y} \in \Re^{4096}$

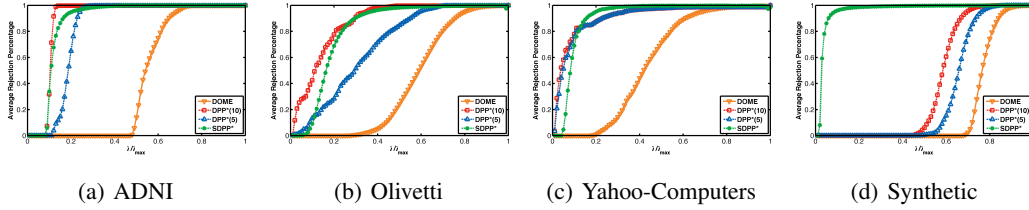

| (a) ADNI | (b) Olivetti | (c) Yahoo-Computers | (d) Synthetic |

Figure 2: Comparison of DPP*/SDPP* rules and Dome on three real data sets, Yahoo computers data set, ADNI data set, Olivetti face data set and one synthetic data set.

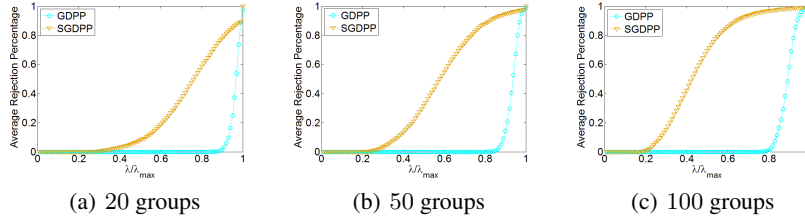

| (a) 20 groups | (b) 50 groups | (c) 100 groups |

Figure 3: Performance of GDPP and SGDPP applied to three synthetic data sets.

and the data matrix $\mathbf{X} \in \Re^{4096 \times 399}$ is constructed by the left ones. **e)** The Yahoo data sets include 11 top-level categories such as Computers, Education, Health, Recreation, and Science etc. Each category is further divided into a set of subcategories. Each time, we construct a balanced binary classification data set from the topic of Computers. We choose samples from one subcategory as the positive class and randomly sample an equal number of samples from the rest of subcategories as the negative class. The size of the data matrix is $876 \times 25259$ and the response vector is the binary label of the samples. **f)** For the synthetic data set $\mathbf{X} \in \Re^{100 \times 5000}$ and the response vector $\mathbf{y} \in \Re^{100}$, all of the entries are i.i.d. by a standard Gaussian.

The average $\lambda_{max}$ of the above three data sets are $0.7273$, $0.989$, $0.914$, and $0.371$ respectively. The predictors in ADNI, Yahoo-Computers and Olivetti data sets are highly correlated as indicated by the average $\lambda_{max}$. In contrast with the real data sets, the average $\lambda_{max}$ of the synthetic data is small. As noted in [26, 25], Dome is very effective in discarding inactive features when $\lambda_{max}$ is large. From Fig. 2, we observe that Dome performs much better on the real data sets compared to the synthetic data. However, the proposed rules are able to identify far more inactive features than Dome on both real and synthetic data, even for the cases in which $\lambda_{max}$ is small.

### 4.2 GDPPs for the Group Lasso Problem

We apply GDPPs to three synthetic data sets. The entries of data matrix $\mathbf{X} \in \Re^{100 \times 1000}$ and the response vector $\mathbf{y}$ are generated i.i.d. from the standard Gaussian distribution. For each of the cases, we randomly divided $\mathbf{X}$ into 20, 50, and 100 groups. We compare the performance of GDPP and SGDPP along a sequence of 100 parameter values equally spaced on the $\lambda/\lambda_{max}$ scale. We repeat the above procedure 100 times for each of the cases and report the average performance. The average $\lambda_{max}$ values are $0.136$, $0.167$, and $0.219$ respectively. As shown in Fig. 3, it is expected that SGDPP significantly outperforms GDPP which only makes use of the information of the dual optimal solution at a single point. For more discussions, please refer to the supplement.

## 5 Conclusion

In this paper, we develop new screening rules for the Lasso problem by making use of the nonexpansiveness of the projection operator with respect to a closed convex set. Our new methods, i.e., DPP rules, are able to effectively identify inactive predictors of the Lasso problem, thus greatly reducing the size of the optimization problem. Moreover, we further improve DPP rules and propose the enhanced DPP rules, that is, the DPP* rules, which are even more effective in discarding inactive predictors than DPP rules. The idea of DPP and DPP* rules can be easily generalized to screen the inactive groups of the group Lasso problem. Extensive experiments on both synthetic and real data demonstrate the effectiveness of the proposed rules. Moreover, DPP and DPP* rules can be combined with any Lasso solver as a speedup tool. In the future, we plan to generalize our idea to other sparse formulations consisting of more general structured sparse penalties, e.g., tree/graph Lasso.

## Acknowledgments

This work was supported in part by NIH (LM010730) and NSF (IIS-0953662, CCF-1025177).

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
