[Supplementary Material]

# Lasso Screening Rules via Dual Polytope Projection (Supplemental Material)

## 1 Deviation of the Dual Problem of Standard Lasso

### 1.1 Dual Formulation

Assuming the data matrix is $\mathbf{X} \in \Re^{N \times p}$, the standard Lasso problem is given by:

$$\inf_{\beta \in \Re^p} \frac{1}{2} \|\mathbf{y} - \mathbf{X}\beta\|_2^2 + \lambda\|\beta\|_1 \tag{1}$$

For completeness, we give a detailed deviation of the dual formulation of (1) in this section. Note that problem (1) has no constraints. Therefore the dual problem is trivial and useless. A common trick [3] is to introduce a new set of variables $\mathbf{z} = \mathbf{y} - \mathbf{X}\beta$ such that problem (1) becomes:

$$\inf_{\beta} \quad \frac{1}{2}\|\mathbf{z}\|_2^2 + \lambda\|\beta\|_1 \tag{2}$$
$$\text{subject to} \quad \mathbf{z} = \mathbf{y} - \mathbf{X}\beta$$

By introducing the dual variables $\eta \in \Re^N$, we get the Lagrangian of problem (2):

$$L(\beta, \mathbf{z}, \eta) = \frac{1}{2}\|\mathbf{z}\|_2^2 + \lambda\|\beta\|_1 + \eta^T \cdot (\mathbf{y} - \mathbf{X}\beta - \mathbf{z}) \tag{3}$$

For the Lagrangian, the primal variables are $\beta$ and $\mathbf{z}$. And the dual function $g(\eta)$ is:

$$g(\eta) = \inf_{\beta, \mathbf{z}} L(\beta, \mathbf{z}, \eta) = \eta^T \mathbf{y} + \inf_{\beta}(-\eta^T \mathbf{X}\beta + \lambda\|\beta\|_1) + \inf_{\mathbf{z}} \left(\frac{1}{2}\|\mathbf{z}\|_2^2 - \eta^T \mathbf{z}\right) \tag{4}$$

In order to get $g(\eta)$, we need to solve the following two optimization problems.

$$\inf_{\beta} -\eta^T \mathbf{X}\beta + \lambda\|\beta\|_1 \tag{5}$$

and

$$\inf_{\mathbf{z}} \frac{1}{2}\|\mathbf{z}\|_2^2 - \eta^T \mathbf{z} \tag{6}$$

Let us first consider problem (5). Denote the objective function of problem (5) as

$$f_1(\beta) = -\eta^T \mathbf{X}\beta + \lambda\|\beta\|_1. \tag{7}$$

$f_1(\beta)$ is convex but not smooth. Therefore let us consider its subgradient

$$\partial f_1(\beta) = -\mathbf{X}^T \eta + \lambda\mathbf{v}$$

in which $\|\mathbf{v}\|_\infty \leq 1$ and $\mathbf{v}^{\mathbf{T}}\beta = \|\beta\|_1$, i.e., $\mathbf{v}$ is the subgradient of $\|\beta\|_1$.

The necessary condition for $f_1$ to attain an optimum is

$$\exists\, \beta', \text{ such that } 0 \in \partial f_1(\beta') = \{-\mathbf{X}^T \eta + \lambda\mathbf{v}'\}$$

where $\mathbf{v}' \in \partial\|\beta'\|_1$. In other words, $\beta', \mathbf{v}'$ should satisfy

$$\mathbf{v}' = \frac{\mathbf{X}^T\eta}{\lambda}, \|\mathbf{v}'\|_\infty \leq 1, {\mathbf{v}'}^T\beta' = \|\beta'\|_1$$

which is equivalent to

$$|\mathbf{x}_i^T\eta| \leq \lambda, i = 1, 2, \ldots, p. \tag{8}$$

Then we plug $\mathbf{v}' = \frac{\mathbf{X}^T\eta}{\lambda}$ and ${\mathbf{v}'}^T\beta' = \|\beta'\|_1$ into Eq. (7):

$$f_1(\beta') = \inf_\beta f_1(\beta) = -\eta^T\mathbf{X}\beta' + \lambda\left(\frac{\mathbf{X}^T\eta}{\lambda}\right)^T\beta' = 0 \tag{9}$$

Therefore, the optimum value of problem (5) is 0.

Next, let us consider problem (6). Denote the objective function of problem (6) as $f_2(\mathbf{z})$. Let us rewrite $f_2(\mathbf{z})$ as:

$$f_2(\mathbf{z}) = \frac{1}{2}(\|\mathbf{z} - \eta\|_2^2 - \|\eta\|_2^2) \tag{10}$$

Clearly,

$$\mathbf{z}' = \operatorname*{argmin}_{\mathbf{z}} f_2(\mathbf{z}) = \eta$$

and

$$\inf_{\mathbf{z}} f_2(\mathbf{z}) = -\frac{1}{2}\|\eta\|_2^2$$

Combining everything above, we get the dual problem:

$$\sup_\eta \quad g(\eta) = \eta^T\mathbf{y} - \frac{1}{2}\|\eta\|_2^2 \tag{11}$$
$$\text{subject to} \quad |\mathbf{x}_i^T\eta| \leq \lambda, \ i = 1, 2, \ldots, p$$

which is equivalent to

$$\sup_\eta \quad g(\eta) = \frac{1}{2}\|\mathbf{y}\|_2^2 - \frac{1}{2}\|\eta - \mathbf{y}\|_2^2 \tag{12}$$
$$\text{subject to} \quad |\mathbf{x}_i^T\eta| \leq \lambda, \ i = 1, 2, \ldots, p$$

By a simple re-scaling of the dual variables $\eta$, i.e., let $\theta = \frac{\eta}{\lambda}$, problem (12) transforms to:

$$\sup_\theta \quad g(\theta) = \frac{1}{2}\|\mathbf{y}\|_2^2 - \frac{\lambda^2}{2}\|\theta - \frac{\mathbf{y}}{\lambda}\|_2^2 \tag{13}$$
$$\text{subject to} \quad |\mathbf{x}_i^T\theta| \leq 1, \ i = 1, 2, \ldots, p$$

## 1.2  Relationship Between The Primal And Dual Variables

Problem (2) is clearly convex and its constraints are all affine. By Slater's condition, as long as problem (2) is feasible we will have strong duality. Denote $\beta^*$, $\mathbf{z}^*$ and $\theta^*$ as optimal primal and dual variables. The Lagrangian is

$$L(\beta, \mathbf{z}, \theta) = \frac{1}{2}\|\mathbf{z}\|_2^2 + \lambda\|\beta\|_1 + \lambda\theta^T \cdot (\mathbf{y} - \mathbf{X}\beta - \mathbf{z}) \tag{14}$$

From the KKT condition, we have

$$0 \in \partial_\beta L(\beta^*, \mathbf{z}^*, \theta^*) = -\lambda\mathbf{X}^T\theta^* + \lambda\mathbf{v}, \text{ in which } \|\mathbf{v}\|_\infty \leq 1 \text{ and } \mathbf{v}^T\beta^* = \|\beta^*\|_1 \tag{15}$$

$$\nabla_{\mathbf{z}} L(\beta^*, \mathbf{z}^*, \theta^*) = \mathbf{z}^* - \lambda\theta^* = 0 \tag{16}$$

$$\nabla_\theta L(\beta^*, \mathbf{z}^*, \theta^*) = \lambda(\mathbf{y} - \mathbf{X}\beta^* - \mathbf{z}^*) = 0 \tag{17}$$

From Eq. (16) and (17), we have:

$$\mathbf{y} = \mathbf{X}\beta^* + \lambda\theta^* \tag{18}$$

From Eq. (15), we know there exists $\mathbf{v}^* \in \partial\|\beta^*\|_1$ such that

$$\mathbf{X}^T\theta^* = \mathbf{v}^*, \ \|\mathbf{v}^*\|_\infty \leq 1 \text{ and } (\mathbf{v}^*)^T\beta^* = \|\beta^*\|_1$$

which is equivalent to

$$|\mathbf{x}_i^T\theta^*| \leq 1, i = 1, 2, \ldots, p, \text{ and } (\theta^*)^T\mathbf{X}\beta^* = \|\beta^*\|_1 \tag{19}$$

From Eq. (19), it is easy to conclude:

$$(\theta^*)^T\mathbf{x}_i \in \begin{cases} \text{sign}(\beta)_i^* \text{ if } \beta_i^* \neq 0 \\ [-1, 1] \text{ if } \beta_i^* = 0 \end{cases} \tag{20}$$

## 2  Lemmas A and B

**Lemma A.** *[1] Let $C$ be a convex set and $P_C(\cdot)$ be the projection operator which projects an arbitrary point onto $C$. Suppose $\mathbf{w}_0 \notin C$ and $\bar{\mathbf{w}} = P_C(\mathbf{w}_0)$ be the projection of $\mathbf{w}_0$ onto $C$. Then for $t \geq 0$, the projection of $\mathbf{w}(t) = t\mathbf{w}_0 + (1-t)\bar{\mathbf{w}}$ coincides with $\bar{\mathbf{w}}$, i.e. $P_C(\mathbf{w}(t)) = \bar{\mathbf{w}}$.*

**Lemma B.** *Given a data matrix $\mathbf{X} = [\mathbf{x}_1, \ldots, \mathbf{x}_p]$, where each column $x_i \in \Re^N$. Let $F = \{\theta : |\mathbf{x}_i^T\theta| \leq 1, i = 1, \ldots, p\}$. Then, for an arbitrary nonzero vector $\mathbf{y} \in \Re^N$, we have*

$$P_F(\mathbf{y}/\lambda_{max} + t\mathbf{v}_1) = \mathbf{y}/\lambda_{max}, \ \forall t \geq 0, \tag{21}$$

*where $\mathbf{v}_1 = \text{sign}(\mathbf{x}_*^T\mathbf{y})\mathbf{x}_*$, $\mathbf{x}_* := \text{argmax}_{\mathbf{x}_i}|\mathbf{x}_i^T\mathbf{y}|$, and $\lambda_{max} = \max_i |\mathbf{x}_i^T\mathbf{y}|$.*

Before we prove Lemma B, let us cite a general result as follows.

**Theorem A.** *[1] Let $C$ be a nonempty closed convex subset of a real Hilbert space $\mathcal{H}$. Then, for every $\mathbf{w} \in \mathcal{H}$ and $\overline{\mathbf{w}} \in C$, we have*

$$\overline{\mathbf{w}} = P_C(\mathbf{w}) \Leftrightarrow \langle \mathbf{w} - \overline{\mathbf{w}}, \mathbf{v} - \overline{\mathbf{w}} \rangle \leq 0, \ \forall \mathbf{v} \in C. \tag{22}$$

Now, let us prove Lemma B.

*Proof.* For notational simplicity, let $\theta(t) = \mathbf{y}/\lambda_{max} + t\mathbf{v}_1$. According to the definition, it is easy to see that

$$\mathbf{y}/\lambda_{max} \in F \text{ and } \mathbf{v}_1^T(\mathbf{y}/\lambda_{max}) = 1.$$

Consider the closed half space $H(\mathbf{v}_1)_- = \{\theta : \mathbf{v}_1^T\theta \leq 1\}$. For any $\theta \in H(\mathbf{v}_1)_-$, we have

$$\langle \mathbf{v}_1, \theta - \mathbf{y}/\lambda_{max} \rangle \leq 0, \tag{23}$$

which results in

$$\langle \mathbf{y}/\lambda_{max} + t\mathbf{v}_1 - \mathbf{y}/\lambda_{max}, \theta - \mathbf{y}/\lambda_{max} \rangle \leq 0, \ \forall t \geq 0. \tag{24}$$

Clearly, Eq. (24) is equivalent to

$$\langle \theta(t) - \mathbf{y}/\lambda_{max}, \theta - \mathbf{y}/\lambda_{max} \rangle \leq 0, \ \forall \theta \in H(\mathbf{v}_1)_-, \ t \geq 0. \tag{25}$$

Moreover, we can observe that $F \subset H(\mathbf{v}_1)_-$. Thus, in view of Theorem A, we have

$$P_F(\theta(t)) = \mathbf{y}/\lambda_{max}, \ \forall t \geq 0,$$

which completes the proof.

$\square$

# 3 Deviation of the Dual Problem of Group Lasso

## 3.1 Dual Formulation

Assuming the data matrix is $\mathbf{X}_g \in \Re^{N \times n_g}$ and $p = \sum_{g=1}^{G} n_g$, the group Lasso problem is given by:

$$\inf_{\beta \in \Re^p} \frac{1}{2} \|\mathbf{y} - \sum_{g=1}^{G} \mathbf{X}_g \beta_g\|_2^2 + \lambda \sum_{g=1}^{G} \sqrt{n_g} \|\beta_g\|_2 \tag{26}$$

Let $\mathbf{z} = \mathbf{y} - \sum_{g=1}^{G} \mathbf{X}_g \beta_g$ and problem (26) becomes:

$$\inf_{\beta} \quad \frac{1}{2} \|\mathbf{z}\|_2^2 + \lambda \sum_{g=1}^{G} \sqrt{n_g} \|\beta_g\|_2 \tag{27}$$

$$\text{subject to} \quad \mathbf{z} = \mathbf{y} - \sum_{g=1}^{G} \mathbf{X}_g \beta_g$$

By introducing the dual variables $\eta \in \Re^N$, the Lagrangian of problem (27) is:

$$L(\beta, \mathbf{z}, \eta) = \frac{1}{2} \|\mathbf{z}\|_2^2 + \lambda \sum_{g=1}^{G} \sqrt{n_g} \|\beta_g\|_2 + \eta^T \cdot (\mathbf{y} - \sum_{g=1}^{G} \mathbf{X}_g \beta_g - \mathbf{z}) \tag{28}$$

and the dual function $g(\eta)$ is:

$$g(\eta) = \inf_{\beta, \mathbf{z}} L(\beta, \mathbf{z}, \eta) = \eta^T \mathbf{y} + \inf_{\beta} \left( -\eta^T \sum_{g=1}^{G} \mathbf{X}_g \beta_g + \lambda \sum_{g=1}^{G} \sqrt{n_g} \|\beta_g\|_2 \right) + \inf_{\mathbf{z}} \left( \frac{1}{2} \|\mathbf{z}\|_2^2 - \eta^T \mathbf{z} \right) \tag{29}$$

In order to get $g(\eta)$, let us solve the following two optimization problems.

$$\inf_{\beta} -\eta^T \sum_{g=1}^{G} \mathbf{X}_g \beta_g + \lambda \sum_{g=1}^{G} \sqrt{n_g} \|\beta_g\|_2 \tag{30}$$

and

$$\inf_{\mathbf{z}} \frac{1}{2} \|\mathbf{z}\|_2^2 - \eta^T \mathbf{z} \tag{31}$$

Let us first consider problem (30). Denote the objective function of problem (30) as

$$\hat{f}(\beta) = -\eta^T \sum_{g=1}^{G} \mathbf{X}_g \beta_g + \lambda \sum_{g=1}^{G} \sqrt{n_g} \|\beta_g\|_2 \tag{32}$$

Let

$$\hat{f}_g(\beta_g) = -\eta^T \mathbf{X}_g \beta_g + \lambda \sqrt{n_g} \|\beta_g\|_2, \qquad g = 1, 2, \ldots, G$$

then we can split problem (30) into a set of subproblems. Clearly $\hat{f}_g(\beta_g)$ is convex but not smooth because it has a singular point at $0$. Consider the subgradient of $\hat{f}_g$,

$$\partial \hat{f}_g(\beta_g) = -\mathbf{X}_g^T \eta + \lambda \sqrt{n_g} \mathbf{v}_g, \qquad g = 1, 2, \ldots, G$$

where $\mathbf{v}_g$ is the subgradient of $\|\beta_g\|_2$:

$$\mathbf{v}_g \in \begin{cases} \frac{\beta_g}{\|\beta_g\|_2} & \text{if } \beta_g \neq 0 \\ \mathbf{u}, \|\mathbf{u}\|_2 \leq 1 & \text{if } \beta_g = 0 \end{cases} \tag{33}$$

Let $\beta_g'$ be the optimal solution of $\hat{f}_g$, then $\beta_g'$ satisfy

$$\exists \mathbf{v}_g' \in \partial \|\beta_g'\|_2, \quad -\mathbf{X}_g^T \eta + \lambda \sqrt{n_g} \mathbf{v}_g' = 0.$$

If $\beta'_g = 0$, clearly, $\hat{f}_g(\beta'_g) = 0$. Otherwise, since $\lambda\sqrt{n_g}\mathbf{v}'_g = \mathbf{X}_g^T\eta$ and $\mathbf{v}'_g = \frac{\beta'_g}{\|\beta'_g\|_2}$, we have

$$\hat{f}_g(\beta'_g) = -\lambda\sqrt{n_g}\frac{(\beta'_g)^T}{\|\beta'_g\|_2}\beta'_g + \lambda\sqrt{n_g}\|\beta'_g\|_2 = 0.$$

All together, we can conclude the

$$\inf_{\beta_g} \hat{f}_g(\beta_g) = 0, \quad g = 1, 2, \ldots, G$$

and thus

$$\inf_{\beta} \hat{f}(\beta) = \inf_{\beta} \sum_{g=1}^{G} \hat{f}_g(\beta_g) = \sum_{g=1}^{G} \inf_{\beta_g} \hat{f}_g(\beta_g) = 0.$$

The second equality is due to the fact that $\beta_g$'s are independent.

Note, from Eq. (33), it is easy to see $\|\mathbf{v}_g\|_2 \leq 1$. Since $\lambda\sqrt{n_g}\mathbf{v}'_g = \mathbf{X}_g^T\eta$, we get a constraint on $\eta$, i.e., $\eta$ should satisfy:

$$\|\mathbf{X}_g^T\eta\|_2 \leq \lambda\sqrt{n_g}, \qquad g = 1, 2, \ldots, G.$$

Next, let us consider problem (31). Since problem (31) is exactly the same as problem (6), we conclude:

$$\mathbf{z}' = \operatorname*{argmin}_{\mathbf{z}} \frac{1}{2}\|\mathbf{z}\|_2^2 - \eta^T\mathbf{z} = \eta$$

and

$$\inf_{\mathbf{z}} \frac{1}{2}\|\mathbf{z}\|_2^2 - \eta^T\mathbf{z} = -\frac{1}{2}\|\eta\|_2^2$$

Therefore the dual function $g(\eta)$ is:

$$g(\eta) = \eta^T\mathbf{y} - \frac{1}{2}\|\eta\|_2^2.$$

Combining everything above, we get the dual formulation of the group Lasso:

$$\sup_{\eta} \quad g(\eta) = \eta^T\mathbf{y} - \frac{1}{2}\|\eta\|_2^2 \tag{34}$$
$$\text{subject to} \quad \|\mathbf{X}_g^T\eta\|_2 \leq \lambda\sqrt{n_g}, \, g = 1, 2, \ldots, G$$

which is equivalent to

$$\sup_{\eta} \quad g(\eta) = \frac{1}{2}\|\mathbf{y}\|_2^2 - \frac{1}{2}\|\eta - \mathbf{y}\|_2^2 \tag{35}$$
$$\text{subject to} \quad \|\mathbf{X}_g^T\eta\|_2 \leq \lambda\sqrt{n_g}, \, g = 1, 2, \ldots, G$$

By a simple re-scaling of the dual variables $\eta$, i.e., let $\theta = \frac{\eta}{\lambda}$, problem (35) transforms to:

$$\sup_{\theta} \quad g(\theta) = \frac{1}{2}\|\mathbf{y}\|_2^2 - \frac{\lambda^2}{2}\|\theta - \frac{\mathbf{y}}{\lambda}\|_2^2 \tag{36}$$
$$\text{subject to} \quad \|\mathbf{X}_g^T\theta\|_2 \leq \sqrt{n_g}, \, g = 1, 2, \ldots, G$$

## 3.2 Relationship Between The Primal And Dual Variables

Clearly, problem (27) is convex and its constraints are all affine. By Slater's condition, as long as problem (27) is feasible we will have strong duality. Denote $\beta^*$, $\mathbf{z}^*$ and $\theta^*$ as optimal primal and dual variables. The Lagrangian is

$$L(\beta, \mathbf{z}, \theta) = \frac{1}{2}\|\mathbf{z}\|_2^2 + \lambda\sum_{g=1}^{G}\sqrt{n_g}\|\beta_g\|_2 + \lambda\theta^T \cdot (\mathbf{y} - \sum_{g=1}^{G}\mathbf{X}_g\beta_g - \mathbf{z}) \tag{37}$$

From the KKT condition, we have

$$0 \in \partial_{\beta_g} L(\beta^*, \mathbf{z}^*, \theta^*) = -\lambda \mathbf{X}_g^T \theta^* + \lambda \sqrt{n_g} \mathbf{v}_g, \text{ in which } \mathbf{v}_g \in \partial \|\beta_g^*\|_2, \quad g = 1, 2, \ldots, G \tag{38}$$

$$\nabla_{\mathbf{z}} L(\beta^*, \mathbf{z}^*, \theta^*) = \mathbf{z}^* - \lambda \theta^* = 0 \tag{39}$$

$$\nabla_\theta L(\beta^*, \mathbf{z}^*, \theta^*) = \lambda \cdot (\mathbf{y} - \sum_{g=1}^{G} \mathbf{X}_g \beta_g^* - \mathbf{z}^*) = 0 \tag{40}$$

From Eq. (39) and (40), we have:

$$\mathbf{y} = \sum_{g=1}^{G} \mathbf{X}_g \beta_g^* + \lambda \theta^* \tag{41}$$

From Eq. (38), we know there exists $\mathbf{v}_g' \in \partial \|\beta_g^*\|_2$ such that

$$\mathbf{X}_g^T \theta^* = \sqrt{n_g} \mathbf{v}_g'$$

and

$$\mathbf{v}_g' \in \begin{cases} \frac{\beta_g^*}{\|\beta_g^*\|_2} & \text{if } \beta_g^* \neq 0 \\ \mathbf{u}, \|\mathbf{u}\|_2 \leq 1 & \text{if } \beta_g^* = 0 \end{cases}$$

Then the following holds:

$$\mathbf{X}_g^T \theta^* \in \begin{cases} \sqrt{n_g} \frac{\beta_g^*}{\|\beta_g^*\|_2} & \text{if } \beta_g^* \neq 0 \\ \sqrt{n_g} \mathbf{u}, \|\mathbf{u}\|_2 \leq 1 & \text{if } \beta_g^* = 0 \end{cases} \tag{42}$$

for $g = 1, 2, \ldots, G$. Clearly, if $\|\mathbf{X}_g^T \theta^*\|_2 < \sqrt{n_g}$, we can conclude $\beta_g^* = 0$.

## 4 Proof of Theorem 8

*Proof.* From the KKT conditions in Eq. (42), we know

$$\|\mathbf{X}_g^T \theta^*(\lambda'')\|_2 < \sqrt{n_g} \Rightarrow \beta_g^*(\lambda'') = 0.$$

By the dual problem (36), $\theta^*(\lambda)$ is the projection of $\frac{\mathbf{y}}{\lambda}$ onto the feasible set which is closed and convex. Note, the feasible set is in fact the intersection of ellipsoids:

$$\{\theta \colon \|\mathbf{X}_g^T \theta\|_2 \leq \sqrt{n_g}\}, g = 1, 2, \ldots, G.$$

According to the projection theorem [2] for closed convex sets, $\theta^*(\lambda)$ is continuous and nonexpansive, i.e.,

$$\|\theta^*(\lambda'') - \theta^*(\lambda')\|_2 \leq \left\| \frac{\mathbf{y}}{\lambda''} - \frac{\mathbf{y}}{\lambda'} \right\|_2 = \|\mathbf{y}\|_2 \left| \frac{1}{\lambda''} - \frac{1}{\lambda'} \right| \tag{43}$$

Then

$$\|\mathbf{X}_g^T \theta^*(\lambda'')\|_2 \leq \|\mathbf{X}_g^T \theta^*(\lambda'') - \mathbf{X}_g^T \theta^*(\lambda')\|_2 + \|\mathbf{X}_g^T \theta^*(\lambda')\|_2 \tag{44}$$

$$< \|\mathbf{X}_g\|_2 \|(\theta^*(\lambda'') - \theta^*(\lambda'))\|_2 + \sqrt{n_g} - \|\mathbf{X}_g\|_F \|\mathbf{y}\|_2 \left| \frac{1}{\lambda'} - \frac{1}{\lambda''} \right|$$

$$\leq \|\mathbf{X}_g\|_F \|\mathbf{y}\|_2 \left| \frac{1}{\lambda''} - \frac{1}{\lambda'} \right| + \sqrt{n_g} - \|\mathbf{X}_g\|_F \|\mathbf{y}\|_2 \left| \frac{1}{\lambda'} - \frac{1}{\lambda''} \right| = \sqrt{n_g}$$

which completes the proof.

We use the fact that $\|\mathbf{X}_g\|_2 \leq \|\mathbf{X}_g\|_F$ in the last inequality of Eq. (44). The subscript $\|\cdot\|_F$ denotes the Frobenius norm. $\square$

# 5 Lemma C

**Lemma C.** *Given a data matrix $\mathbf{X} = [\mathbf{X}_1, \ldots, \mathbf{X}_G]$, where each $\mathbf{X}_g \in \Re^{N \times n_g}$. Let $F = \{\theta : \|\mathbf{X}_g^T \theta\|_2 \leq \sqrt{n_g}, g = 1, \ldots, G\}$. Then, for an arbitrary nonzero vector $\mathbf{y} \in \Re^N$, we have*

$$P_F(\mathbf{y}/\lambda_{max} + t\mathbf{v}_1) = \mathbf{y}/\lambda_{max}, \ \forall t \geq 0, \tag{45}$$

*where $\mathbf{v}_1 = \mathbf{X}_* \mathbf{X}_*^T \mathbf{y}$, $\mathbf{X}_* := \operatorname{argmax}_{\mathbf{X}_g} \|\mathbf{X}_g^T \mathbf{y}\|_2 / \sqrt{n_g}$, and $\lambda_{max} = \max_g \|\mathbf{X}_g^T \mathbf{y}\|_2 / \sqrt{n_g}$.*

*Proof.* For simplicity, let $\theta(t) = \mathbf{y}/\lambda_{max} + t\mathbf{v}_1$. From the definition, it is easy to see that

$$\frac{\|\mathbf{X}_*^T \mathbf{y}\|_2}{\sqrt{n_*}} = \lambda_{max}, \tag{46}$$

where $n_*$ is the number of columns of $\mathbf{X}_*$.

Let $\mathbf{u} = \mathbf{v}_1/(\lambda_{max}\sqrt{n_*})$ and $H(\mathbf{u})_- = \{\theta : \mathbf{u}^T \theta \leq \sqrt{n_*}\}$. We can see that

$$\mathbf{u}^T(\mathbf{y}/\lambda_{max}) = \frac{1}{\sqrt{n_*}} \left\| \frac{\mathbf{X}_*^T \mathbf{y}}{\lambda_{max}} \right\|_2^2 = \sqrt{n_*}, \tag{47}$$

and thus

$$\langle \mathbf{u}, \theta - \mathbf{y}/\lambda_{max} \rangle \leq 0, \ \ \forall \theta \in H(\mathbf{u})_-. \tag{48}$$

According to the definition of $\mathbf{u}$, we also have

$$\langle \mathbf{v}_1, \theta - \mathbf{y}/\lambda_{max} \rangle \leq 0, \ \ \forall \theta \in H(\mathbf{u})_-. \tag{49}$$

Let $F_* = \{\theta : \|\mathbf{X}_*^T \theta\|_2 \leq \sqrt{n_*}\}$. It is easy to see that $F \subset F_*$. Moreover, we claim that $F_* \subset H(\mathbf{u})_-$. Suppose $\theta \in F_*$, then

$$\mathbf{u}^T \theta = \frac{1}{\lambda_{max}\sqrt{n_*}} \langle \mathbf{X}_* \mathbf{X}_*^T \mathbf{y}, \theta \rangle = \frac{1}{\lambda_{max}\sqrt{n_*}} \langle \mathbf{X}_*^T \mathbf{y}, \mathbf{X}_*^T \theta \rangle \leq \frac{1}{\lambda_{max}\sqrt{n_*}} \|\mathbf{X}_*^T \mathbf{y}\|_2 \|\mathbf{X}_*^T \theta\|_2 \leq \sqrt{n_*}. \tag{50}$$

Therefore, we can see that $\theta \in H(\mathbf{u})_-$ and thus $F_* \subset H(\mathbf{u})_-$. In summary, we have the following statement holds:

$$F \subset F_* \subset H(\mathbf{u})_-. \tag{51}$$

In view of (49) and (51), we can conclude that

$$\langle \mathbf{v}_1, \theta - \mathbf{y}/\lambda_{max} \rangle \leq 0, \ \ \forall \theta \in F, \tag{52}$$

and thus

$$\langle \theta(t) - \mathbf{y}/\lambda_{max}, \theta - \mathbf{y}/\lambda_{max} \rangle \leq 0, \ \ \forall \theta \in F, t \geq 0. \tag{53}$$

Clearly, in view of Theorem A and (53), the statement follows directly. $\square$

# 6 Additional Empirical Results

In this section we report additional experimental results.

## 6.1 Synthetic Data Sets

We generate three synthetic data sets with different dimensions. For each of the cases, the entries of data matrix $\mathbf{X}$ and response vector $\mathbf{y}$ are independent identically distributed by a standard Gaussian. Each data matrix contains 100 samples with $p = 50, 500$, and $5000$ respectively. For each case, once we generate the data matrix $\mathbf{X}$, we compare the performance of DPP* rules with Dome along a sequence of 100 parameter values equally spaced on the $\lambda/\lambda_{max}$ scale. Then we repeat the procedure 100 times and report the average performance of each rule.

The three subfigures of Fig. 1 correspond to the three different design matrices $\mathbf{X}$ and the average $\lambda_{max}$ is 0.249, 0.315 and 0.371 respectively. As shown in Fig. 1, the performance of DPP* is comparable to Dome but all the other DPP* rules significantly outperform Dome. In contrast to Dome which performs better with larger $\lambda_{max}$ [4], DPP* rules exhibit stronger capability in discarding inactive predictors when $\lambda_{max}$ is small. The geometric intuition behind this observation is due to the fact that the sparser the predictors distribute over the unit ball, the longer the line segment of the regularization path is. If the length of the line segment of the regularization path is larger, the first few breakpoints may correspond to very small $\lambda$ values.

(a) $\mathbf{X} \in \Re^{100 \times 50}$      (b) $\mathbf{X} \in \Re^{100 \times 500}$      (c) $\mathbf{X} \in \Re^{100 \times 5000}$

Figure 1: Comparison of DPP* rules and Dome on three synthetic datasets. Each column corresponds to each of the three synthetic data sets with different dimensions.

## 6.2 Discussions of GDPP

For the group Lasso problem, the feasible set of its dual variables is the intersection of ellipsoids and is thus no longer a polytope. As a consequence, the path of the optimal solution is no longer piecewise linear. Due to this fact, it is more complicated to characterize the path and find the breakpoints where groups of predictors enter or leave the active set. However, if there are efficient algorithms which can find the breakpoints and the corresponding parameters like LARS for Lasso, we can potentially make use of those breakpoints and the associated parameters to construct more effective screening rules based on Theorem 8.