[Reviews · NeurIPS 2013]

Submitted by Assigned_Reviewer_1

In this paper, the authors propose a new screening rule for Lasso. They give some theoretical fundamentals and describe its extension to GroupLasso. The screening technique for Lasso is an important problem for the community. However, to be honest, I am not so familiar with the topic enough to evaluate whether their theoretical results are novel. But at least, I could say that the current paper is somehow unkindly organized for readers. Also, there is a concern that they just give empirical comparison of their algorithm only with a somehow minor existing screening method (Dome).
Summary: In this paper, the authors propose a new screening rule for Lasso. Although I am not so familiar with the topic enough to evaluate whether their theoretical results are novel, at least I could say that the current paper is somehow unkindly organized for readers.

Submitted by Assigned_Reviewer_7

Title:
Lasso Screening Rules via Dual Polytope Projection

Summary:
The paper proposes exact and improved screening rules for solving sparse regression problems by means of dual polytope projections.

Quality:
The paper is relatively complete with motivations, full derivations of algorithms and their proofs from the basics, and an adequate amount of experimental results.
However, the emphasis is on the derivations, and may not be of interest to those readers who wants to see a bigger picture of the what the paper promises for machine learning problems in general.

Originality:
My expertise on this specific problem is limited. With that in mind, the paper seems to take an approach closely related to those of [26] or [27], while it diverges from those in the details, in particular with the sequential extensions.

Clarity:
Overall, the paper clearly presents its goal and the proposal which is the reduction of the active set exactly in Lasso. Its theorems are proved every step of the derivations, mostly in Supplementary material, up until the point where it becomes a bit abstract about the sequential version (SDPP and SGDPP). A summary of algorithm for SDPP/SGDPP would be helpful.
The experiment section is a bit brief: the performances are compared only by the rejection rate and within the group of similar algorithms, with no indication of where Lasso stands among other 'standard' approaches such as SVMs. By the way, labels on the axes in Fig.2-3 are unreadable.

Significance:
The paper has a narrowly defined focus on the specific problem of solving large-scale Lasso problems. As such it seems to suggest a concrete progress on the exact rejection of inactive features without additional costs, and may inspire other related research in general.
Summary: Summary:
From a viewpoint of a reader who has a broad interest in large-scale learning and optimization, the paper is readable and logical, and presents a reasonable amount of contributions although very technical in nature.

Submitted by Assigned_Reviewer_8

The authors propose the use of dual polytope projection as a lasso screening test, and described how ideas in regularization paths and an enhancement DPP* can lead to an improved screening test. They showed how the test can be extended to group lasso, and also demonstrated quite convincingly the utility of their test in experiments.


pros:
- The new dual polytope projection rules work better than previous lasso screening tests such as the dome rule. The authors also manage to demonstrate the utility of using homotopy ideas for picking \lambda, and the improvements given by the enhanced rule DPP*.
- The authors also demonstrate how the screening rules can be extended to group lasso.
- The paper is very well organized and clearly written, with careful experimental evaluations.

cons:
- It is not easy to see how many breakpoints one should solve to get a good screening rule. The authors show DPP2, DPP5, and DPP10 in the experiments, For some problems solving for 5 to 10 breakpoints lead to large improvements in screening effectiveness (Fig 2a,b,c), while for some others they do not (Fig 2d). The authors also did not show how much computational costs solving for these extra breakpoints would incur. It will be good to have some statistics simliar to Table 1 for these real problems.

Summary: The authors propose the use of dual polytope projection as a lasso screening test, and described how ideas in regularization paths and an enhancement DPP* can lead to an improved screening test. They showed how the test can be extended to group lasso, and also demonstrated quite convincingly the utility of their test in experiments.

Submitted by Assigned_Reviewer_10

The authors develop exact screening rules for the Lasso and Group Lasso problems (with quadratic loss). These rules are based on the observation that the dual of these problems is simply the projection of a vector onto a convex polytope. They can be applied in a sequential fashion along a grid of tuning parameter values.

Quality:

This is an excellent paper. The idea is elegant and explained well, and the resulting method appears to be highly effective.

Clarity:

Very good.

Originality:

While the main idea is not greatly different from the SAFE rules, the authors have simplified the idea and made it work better.

Significance:

This seems like work that could be easily built on. While the "exact" aspect of DPP is nice, the strong rules paper makes the point that their rule often just works (even though it requires checking the KKT conditions). It would be helpful to see the strong rule compared in the empirical studies. Of course for the strong rules you'd need to report whether it threw out any variables it should not have (and if so, perhaps show how many false rejections there were). Regardless of the results, DPP is still significant for being the best

Comments:

It is easy to prove that the DPP rule (at lambda_max) is strictly better than the SAFE rule, though I don't think you come out and say this. To compare the two, notice that DPP can be written as

|x_i^T y | < lambda_max - ||x_i|| ||y|| (lamda_max - lambda) / lambda

Comparing the RHS of DPP and SAFE are identical except for two differences. Both of these differences make RHS's DPP larger than SAFE and therefore it can screen strictly more features than SAFE.

Minor comments:
- Equation 8: Fix RHS.
- Equations 21,22: Fix LHS (should be L2 norms)
- Line 346: "projection ratio" should be "rejection ratio" ?
- Line 376: "verfying" typo
- Lines 406-7: highly correlated with the response, you mean?



Summary: A well-written paper that describes a method that beats the state-of-the-art "exact" screening rules for the Lasso and Group Lasso problems.
Author Feedback

Author rebuttal: We thank all of the reviewers for the constructive comments.

Reviewer 1

Q: The authors just give empirical comparison of their algorithm with a somehow minor existing screening method (Dome).

A: As demonstrated in the papers by Xiang et al. (2011) and Xiang et al. (2012), Dome significantly outperforms SAFE in discarding the inactive features among the safe screening methods. Dome is the best safe screening methods according to our tests.

Reviewer 10

Q: It would be helpful to compare strong rule with the proposed method.

A: Thanks for this nice suggestion. In this paper, we focus on the safe screening methods.

Q: Lines 406-407, what do you mean by “highly correlated with the response”?

A: By “highly correlated with the response”, we mean the absolute values of the correlation coefficients between the features and the response vector are large.

Reviewer 7

Q: the paper seems to take an approach closely related to those of [26] or [27], while it diverges from those in the details, in particular with the sequential extensions.

A: All of the work in [26], [27] and this paper are inspired by SAFE rules [9]. The most challenging part in developing screening methods is usually the estimation of the dual optimal solution. The geometric background of the proposed methods for estimating the possible region of the dual optimal solution is totally different from that of [26] and [27]. Our methods rely on the nonexpansiveness of the projection operator defined in an arbitrary Hilbert space and thus can be easily extended to sequential version and group Lasso. The sequential screening and group Lasso screening were not considered in [26] and [27], and it is unclear if such extensions exist.

Reviewer 8

Q: It is not easy to see how many breakpoints one should solve to get a good screening rule.

A: Empirically, DPP 5 or DPP 10 results in very good performance for all real data sets used in this paper.

Q: The authors also did not show how much computational costs solving for these extra breakpoints would incur.

A: Thanks for this nice suggestion. Since we make use of LARS to find the breakpoints, the computational cost is very low especially for the first few breakpoints [roughly O(kNp) where k, N and p are the number of breakpoints, samples and feature dimension respectively].